# *Mec*-Positive *Staphylococcus* Healthcare-Associated Infections Presenting High Transmission Risks for Antimicrobial-Resistant Strains in an Equine Hospital

**DOI:** 10.3390/antibiotics11050621

**Published:** 2022-05-04

**Authors:** Paula Soza-Ossandón, Dácil Rivera, Kasim Allel, Gerardo González-Rocha, Mario Quezada-Aguiluz, Ivan San Martin, Patricia García, Andrea I. Moreno-Switt

**Affiliations:** 1Escuela de Medicina Veterinaria, Facultad de Ecología y Recursos Naturales, Universidad Andres Bello, Santiago 9340000, Chile; paula.soza.ossandon@gmail.com; 2Department of Disease Control, Faculty of Infectious & Tropical Diseases, London School of Hygiene & Tropical Medicine, London WC1E 7HT, UK; k.allel@ucl.ac.uk; 3College of Medicine and Health, University of Exeter, Exeter EX1 2LU, UK; 4Institute for Global Health, University College London, London WC1N 1EH, UK; 5Laboratorio de Investigación en Agentes Antibacterianos, Departamento de Microbiología, Facultad de Ciencias Biológicas, Universidad de Concepción, Concepción P.O. Box C-160, Chile; ggonzal@udec.cl (G.G.-R.); marioquezada@udec.cl (M.Q.-A.); ivansanmartin@udec.cl (I.S.M.); 6Departamento de Medicina Interna, Facultad de Medicina, Universidad de Concepción, Concepción 4030000, Chile; 7Escuela de Medicina, Facultad de Medicina, Pontificia Universidad Católica de Chile, Santiago 8940000, Chile; pgarciacan@uc.cl; 8Escuela de Medicina Veterinaria, Facultad de Agronomía e Ingeniería Forestal, Facultad de Ciencias Biológicas, Facultad de Medicina, Pontificia Universidad Católica de Chile, Santiago 8940000, Chile

**Keywords:** methicillin resistance, one health, antimicrobial-resistance, healthcare-associated infections, *Staphylococcus*

## Abstract

Healthcare-associated infections caused by *Staphylococcus*, particularly *Staphylococcus aureus*, represent a high risk for human and animal health. *Staphylococcus* can be easily transmitted through direct contact with individual carriers or fomites, such as medical and non-medical equipment. The risk increases if *S. aureus* strains carry antibiotic resistance genes and show a phenotypic multidrug resistance behavior. The aim of the study was to identify and characterize methicillin resistant coagulase-positive staphylococci (MRSA) and coagulase-negative staphylococci (MRCoNS) in equine patients and environmental sources in an equine hospital to evaluate the genetic presence of multidrug resistance and to understand the dissemination risks within the hospital setting. We explored 978 samples for MRSA and MRCoNS using Oxacillin Screen Agar in an equine hospital for racehorses in Chile, which included monthly samples (*n* = 61–70) from equine patients (246) and hospital environments (732) in a one-year period. All isolates were PCR-assessed for the presence of methicillin resistance gene *mecA* and/or *mecC*. Additionally, we explored the epidemiological relatedness by Pulsed Field Gel Electrophoresis (PFGE) in MRSA isolates. Phenotypic antibiotic resistance was evaluated using the Kirby-Bauer disk diffusion method. We estimated the unadjusted and adjusted risk of acquiring drug-resistant *Staphylococcus* strains by employing logistic regression analyses. We identified 16 MRSA isolates and 36 MRCoNS isolates. For MRSA, we detected *mec*A and *mec*C in 100% and 87.5 % of the isolates, respectively. For MRCoNS, *mec*A was detected among 94% of the isolates and *mecC* among 86%. MRSA and MRCoNS were isolated from eight and 13 equine patients, respectively, either from colonized areas or compromised wounds. MRSA strains showed six different pulse types (i.e., A1–A3, B1–B2, C) isolated from different highly transited areas of the hospital, suggesting potential transmission risks for other patients and hospital staff. The risk of acquiring drug-resistant *Staphylococcus* species is considerably greater for patients from the surgery, equipment, and exterior areas posing higher transmission risks. Tackling antimicrobial resistance (AMR) using a One Health perspective should be advocated, including a wider control over antimicrobial consumption and reducing the exposure to AMR reservoirs in animals, to avoid cross-transmission of AMR *Staphylococcus* within equine hospitals.

## 1. Introduction

The emergence of healthcare-associated infections (HAI), along with the rise of zoonotic and multidrug-resistant pathogens within hospitals, continues to constitute a great disease burden affecting population health [1,2]. Though interventions in veterinary hospitals have been encouraged [3], developing countries are still behind [4,5]. Drug-resistant bacteria represent a significant challenge for human and veterinary medicine, and environmental health [6], because they can remain for a long period of time in hospital environments facilitating the spread of resistant genes [7]. Furthermore, opportunistic multidrug-resistant (MDR) pathogens have long been considered endemic in veterinary hospital settings [8]. Their presence is considered a cause of major concern due to their innate ability to propagate and the very limited treatment options available to control them [9]. The close link between humans and domestic animals provides opportunity venues for the exchange of microorganisms, including MDR pathogens. For instance, outbreaks linking horses and humans in veterinary hospitals have been reported in different countries, evidencing the risks of a bidirectional spread [10].

Additionally, hospitalized patients are susceptible for HAIs within hospital settings because of the exposure to different antimicrobials and the high probability of having a compromised immune system, while undergoing surgical procedures and/or invasive surgeries [6]. Among the most relevant HAI pathogens in veterinary medicine are methicillin-resistant *Staphylococcus aureus* (MRSA) [11,12,13,14] and methicillin-resistant-coagulase-negative staphylococci (MRCoNS) (e.g., *S. vitulinus, S. sciuri, S. haemolyticus, S. pseudintermedius, S. epidermidis*) [15,16,17,18]. Moodley *et al.* identified persistent clones of *S. vitulinus* and *S. haemolyticus* in horses and human contact surfaces, suggesting a horses-humans exchange of MRCoNS strains through direct contact or indirectly through the use of objects and/or the exposure to contaminated environments [15]. In *Staphylococcus* spp., the acquisition of either *mec*A or *mec*C genes allows the pathogen to develop resistance mechanisms against antibiotics, including penicillins, cephalosporins and carbapenems [15,17,18,19]. MRCoNS may operate as a donor transferring the *mec* genes to *Staphylococcus aureus* constituting a reservoir of resistance genes for MRSA [20,21,22]. Moreover, healthcare staff working in veterinary hospitals, especially those working with horses, are at greater risk of MRSA colonization; evidencing an occupation-related health risk [1,13,23]. Consequently, *S. aureus* is significantly relevant as HAI, due to its dynamic pathogenic capacity of causing minor skin and foreign body infections to even fatal septicemia [15,24,25,26]. Even though MRSA emerged as a human pathogen, it also affects domestic animals, with reports of considerable burden in horses and companion animals within the last years [1,15,24,25,26].

The aim of this study was to identify and characterize MRSA and MRCoNS in equine patients and environmental sources in an equine hospital in order to evaluate the genetic presence of multidrug resistance and to understand the dissemination risks within the hospital setting.

## 2. Results

### 2.1. Mec-Positive Staphylococcus Strains Were Isolated from the Equine Hospital

We collected 978 samples from two different sources: 246 from equine patients and 732 from environmental sources from the hospital. A total of 16 MRSA and 36 MRCoNS were obtained (Table 1 and Table 2). In isolates representing *Mec*-positive *S. aureus* (MRSA), 50% (8/16) were isolated from six equine patients and 50% (8/16) from environmental samples, mostly from common contact surfaces (Table 1 and Table 2). MRSA isolates were found in the stalls of the hospitalization area - mainly used by human-supervised equine patients - classified as a common contact surface. We also isolated MRSA from the stalls of the surgery area and medical and non-medical equipment, such as pitchforks, gastroscope, and waterers (Table 2). Among the 36 isolates of MRCoNS, we found different species, such as *S. sciuris* (*n* = 20), *S. vitulinus* (*n* = 4), and *S. lentus* (*n* = 1). There were 11 additional isolates, the species of which could not be determined and therefore were classified as *Staphylococcus* spp. (Table 3). MRCoNS were mostly isolated from common contact surfaces 61.5% (22/36) and equine patients 38.5% (14/36) (either from colonized areas or compromised wounds).

In general, MRSA and MRCoNS isolation was observed in all areas of the equine veterinary hospital (EVH). Interestingly, 31% of MRSA isolates (5/16) and 36% of MRCoNS (13/36) were obtained from colonized patients. A tendency to a greater isolation of MRSA was observed in spring season, while for MRCoNS it was in winter (Appendix A). MRSA and MRCoNS were detected in 10 out of 12, and in eight out 12 months of this study duration. 

### 2.2. Antimicrobial Resistance Profile of Mec-Positive Staphylococcus 

The disk diffusion test in MRSA isolates found two isolates phenotypically pan-susceptible; one isolate was resistant to penicillin (PEN); one isolate to PEN and oxacillin (OXA); nine isolates were resistant to cefoxitin (FOX), gentamicin (GEN), OXA, PEN and tetracycline (TET); one isolate was resistant to azithromycin (AZM), FOX, ciprofloxacin (CIP), clindamycin (CLI), OXA and PEN; one isolate to FOX, CLI, GEN, OXA, PEN and TET; and one isolate to AZM, FOX, CIP, CLI, GEN, OXA, PEN, and TET. Of these, 75% (12/16), were classified as phenotypically MDR [16]. The antimicrobial drugs generating higher antimicrobial resistance levels among *S. aureus* strains (MRSA) were PEN (88%; 14/16), OXA (81%; 13/16), and FOX (75%; 12/16) (Table 2). 

Regarding MRCoNS, five isolates were phenotypically resistant to FOX - CLI - OXA - PEN, and two isolates were phenotypically resistant to FOX - CLI - GEN - OXA - PEN. All the rest of the MRCoNS showed different antimicrobial resistant profiles, but all of them were primarily resistant to oxacillin (OXA) at 86% (31/36), penicillin (PEN) at 75% (27/36), cefoxitin (FOX) at 72% (26/36), clindamycin (CLI) at 47% (17/36), gentamycin (GEN) at 39% (14/36), (Table 2). Most of them (86%; 31/36) fell into the classification of phenotypical MDR [16] (Table 3). 

### 2.3. mecA and mecC Were Found in MRSA and MRCoNS 

On the 16 MRSA analyzed, only four isolates encoded *mec*A and the remaining 12 isolates encoded for both genes *mec*A and *mec*C (Table 2), suggesting the presence of a possible rare SCC*mec*. These results, together with the clinical relevance of MRSA, led us to further analysis, which consisted of performing additional PCR testing according to Stegger *et al.* to search for sequence variations in the SCC*mec* [27]. We used *mec*A_LGA251_MultiFP and *mec*A_LGA251_RP primers for three isolates selected at random. PCR amplicons of 720-bp were sequenced and the BLAST algorithm yielded positive, with a 100% match, to *S. sciuri* subsp. *carnaticus* SCCmec-mecC region (Accession N° HG515014.1). Also, an identity matrix was performed using Clustal Omega 2.1 revealing a 99.5% of identity in two isolates, and a 97.4% identity with the third isolate (Appendix A, Appendix A). On the 36 MRCoNS, we found that five isolates contained *mec*A*,* 2 isolates contained *mec*C, and 29 isolates contained both (Table 3). 

### 2.4. Diversity of PFGE Profiles in MRSA Strains

We further analyzed the genetic relationships of the 16 MRSA isolates. We found six different PFGE pulse-types which were identified in all hospital areas, including common contact surfaces, colonized patients, and wounds. Pulse-type-A1 contained eight isolates, Pulse-type-A2 contained two isolates, Pulse-type-A3 contained one isolate, Pulse-type-B1 contained two isolates, Pulse-type-B2 contained one, and Pulse-type-C contained two isolates (Table 2; Figure 1). Pulse-types A1, A2, A3, B1 and B2 were isolated from common contact surfaces or patients, and pulse-type C was isolated from a human contact surface (Table 2).

### 2.5. Risk Factors for the Acquisition of Resistance by MRCoNS or either MRSA Strains

We conducted a univariate regression analysis using four different definitions of antimicrobial resistance profiles: (I) MRCoNS, (II) MRSA, (III) either MRCoNS or MRSA, (IV) MRCoNS and MRSA strains simultaneously (Figure 2). Environmental samples from common contact surface, samples collected during winter, and the hospitalization area were significantly related to antimicrobial resistance (*p*-value < 0.1) in the univariate analysis (negatively, positively, positively, and negatively, respectively). Therefore, we incorporated those variables in further multivariate analyses (*p*-value < 0.05) (Appendix A and Appendix A). The risk of acquiring antimicrobial resistant strains is considerably low for patients from the hospitalization area compared to environmental samples from any other areas, in almost every single model (e.g., OR_III_ = 0.03, 95%CI = 0.00,0.31, *p*-value < 0.001). Conversely, those samples collected during the winter were associated with a higher risk of either MRSA or MRCoNS (OR_III_ = 2.62, 95%CI = 1.38,4.98, *p*-value < 0.001). Finally, environmental samples (compared to patient’s) had a protective association against MRCoNS (OR_II_= 0.02, 95%CI = 0.00,0.31, *p*-value < 0.001) and it is similar to patients presenting joint resistant strains for MRSA and MRCoNS (OR_IV_= 0.001, 95%CI = 0.00,0.01, *p*-value < 0.001).

## 3. Discussion

This article explored the presence of MRSA and MRCoNS in equine patients and environmental sources from a Chilean veterinary hospital. The main implications are presented hereinafter.

### 3.1. Hot Spots of Pathogens and Risks for Healthcare-Acquired Infections

We found that 50% (8/16) of the MRSA isolates came from patients (five colonized and three having wound infections). The remaining 50% came from environmental samples, mostly from common contact surfaces. Interestingly, a previous study performed in a veterinary teaching hospital with a small animal, equine and production animal section isolated similar proportions of MRSA in common and human contact surfaces [28]. Since the observed hospital is exclusive for equine patients, we suggest that horses may be acting as a reservoir for MRSA in line with previous literature [1,10]. Even more worrisome, MRSA isolates from surgical site infections in hospitalized patients (which considerably increased their length of stay) might indicate undiagnosed hospital-acquired infections [22,29]. This might lead to higher patient risk for hospital-acquired infections since common contact surfaces account for a direct connection between equine patients and humans (including veterinarians, nurses, students, horsemen, etc.). High traffic surfaces, regardless if they are human or common contact, have also been concluded [28]. The fact that MRSA was isolated in almost every sampled month shows the absence of seasonality. However, the number of isolates was relatively small and only consisted of hospitalized patients (not ambulatory). Future research might be necessary to conclude any seasonal pattern. Nonetheless, it highlights the survival capacity of these microorganisms under the different environmental conditions [30].

Regarding MRCoNS, though a clear majority of environmental isolates (22/36) found in this study belonged to samples taken from common contact surfaces (17/22), a fair number of environmental isolates were found in human contact surfaces (5/22). Horses have been described as reservoirs of *Staphylococcus-*coagulase-positive [21], and they might act as reservoirs in this study as well. However, MRCoNS was isolated from computer keyboards, which makes us think that healthcare staff might contribute to the spread of these microorganisms among the equine veterinary hospital, as previous literature has also suggested [20,31]. We found various MRCoNS isolates in this study, with only four isolates sharing the same characteristics. Perhaps this information might suggest a constant reintroduction and important diversity of the isolates rather than a persistent presence. It is well documented that MRCoNS spread has arisen during the past years, as they have been responsible for hospital-acquired severe infections in immunocompromised patients [1,18,20]. Moreover, MRCoNS takes action as a donor of resistant genes for other bacterial species including *Staphylococcus aureus* [20,21,32]. Previous results have evidenced that almost every surface sampled in this study may be a potential reservoir of MDR genes which could be transferred to other bacteria of major clinical relevance [33]. The hospitalization and surgery areas, together with common contact surfaces, concentrated a vast amount of *Mec*-positive *Staphylococcus* spp. strains (including a wide diversity of subspecies), entailing high transmission risks within the hospital setting. Fragile patients and horsemen, the frequent use of antibiotics, insufficient hygiene practices, and the spread of highly resistant pathogens in hospital wards due to patient transfer and repositioning create a perfect transmission route exposing patients to a greater disease burden [34].

The fact of having found a vast microbial burden among hospital environmental samples, compared to patients’, can be explained by a substantial bacterial bioburden within the hospital environment, probably because stringent hygiene and sanitation are not in place. Even if they were, patients shed microorganisms to their surrounding environment, which together with intrinsic surface contamination and healthcare worker’s bacterial carriage might contribute significantly to the greater bacterial load and subsequent conformation of environmental reservoirs [7]. This results in increased risks of highly pathogenic hospital-acquired infections among patients. Still, the relationship between environmental and patients’ bacterial load is not strictly direct, and environmental-to-patient bacterial dissemination may take longer to colonize and cause infection among the latter subjects.

### 3.2. Staphylococcus aureus Isolates Are Phenotypically MDR and Encode mecA and mecC Genes Suggesting an Unusual SCCmec 

All MRSA and MRCoNS isolates included in the study yielded positive for the *mec* gene, either A, C or both. This information may be sufficient to consider the isolates obtained as MDR [16]. Nonetheless, the Kirby Bauer test was performed to assess the phenotypically resistant behavior of the isolates. The most repeated antibiotics causing resistance were penicillin, oxacillin, cefoxitin, gentamicin and tetracycline; all of which have been previously described [29,35]. 

We only found five phenotypically pan-susceptible isolates: two MRSA and three MRCoNS. This could be explained by regulatory genes which suppress *mec* genes, and the subsequent encoding of PBP2a, in the absence of β-lactam antibiotics [26]. However, further analyses, such as sequencing the whole SCC*mec*, and analyzing regulatory genes are necessary to better understand the phenotype of these isolates. Regarding the molecular analysis, 75% (12/16) yielded positive for both *mec*A and *mec*C genes, and as for the MRCoNS, 80% (29/36) encoded both *mec* genes. We infer that the novel hybrid staphylococcal chromosome cassette (SCC*mec*) could have been transferred to *Staphylococcus aureus*, as previously described by Harrison *et al*. using samples of bovine infections of *S. sciuris* [32]. This SCC*mec* interestingly harbors *mec*A and *mec*C genes, and phylogenetic analysis suggests that this *mec*C gene is closely related to *Staphylococcus aureus*_LGA251_, and the evidence suggests that both genes contribute to phenotypic oxacillin-resistance [32]. Further analysis of the genetic context of the rest of the MRSA and MRCoNS isolates obtained in this study is necessary to understand the structure of their gene cassettes. 

### 3.3. One Major Pulse-Type of MRSA Could Be Identified, Which Was Isolated from Different Sources and in Different Sampling Months

The occurrence of MRSA isolates from the same pulse-type (A1)—obtained from colonized patients, surgical site infections and from the hospital environment, and during different months of sampling—suggest the presence of a permanent isolate circulating across the veterinary hospital [22,36]. Nevertheless, the constant reintroduction of MRSA isolates of the same characteristics cannot be ruled out [36]. The need for awareness and implementation of infection prevention and control measures can be derived from the present study amongst previous literature [22]. This can be achieved through policies such as better hand hygiene and improved cleaning and disinfection techniques for surfaces and equipment [22]. By adopting these kind of interventions, MRSA incidence in European hospitals has decreased significantly [37], proving substantial need to emulate those protocols within other country settings. Since MRSA can affect humans and horses, the presence of these zoonotic pathogens may increase occupational and nosocomial infection risks in equine hospitals in Chile. Therefore, preventing potential MRSA strains and controlling its reservoirs is crucial to prevent outbreaks.

## 4. Materials and Methods

### 4.1. Study Design

A total of 978 samples were obtained from a longitudinal study conducted between July 2015 and June 2016. We collected 246 samples from patients and 732 from environmental sources in a one-year study. Samples were taken monthly from an Equine Veterinary Hospital (EVH) located in a thoroughbred racetrack in a middle-income municipality of Chile’s capital city (Santiago), which has an approximate inpatient population of 75 animals daily. The EVH was divided into five areas regarding environmental sampling: exterior, equipment, proceedings, surgery and hospitalization area, (Figure 2), similar to previous studies [38,39].

Furthermore, sampled surfaces were classified into common contact surfaces (direct contact of animals and humans) (*n* = 396) and human contact surfaces (direct contact of humans, but out of reach of animals) (*n* = 96), which has been previously described elsewhere [28] (Figure 2). Regarding equine patients, at least four samples per patient were obtained. This included colonization areas such as nostrils (both), armpits (both), and wounds, including surgical ones (if any) [36]. 

### 4.2. Sample Procedure

Patient samples were collected using sterile swabs, rubbed in the target area, and put into Stuart transport media (COPANTM, Murrieta, USA). Environmental samples were obtained using a sterile gauze soaked in 90 mL of peptone water (Beckton-Dickinson™, Franklin Lakes, NJ, USA) and rubbed on the surface for 5 min (Appendix A).

### 4.3. Staphylococcus spp. Isolation and Identification

All environmental samples were cultured in peptone water at 37 °C overnight as a pre-enrichment method. Then, we used Oxacillin Screen Agar (ORSA) (OXOID, Hampshire, UK) as the first screening approach [40] (Appendix A). Presumptive colonies were submitted to the microbiology unit of the clinical laboratory service of Red Salud UC for mass spectrometry by MALDI-TOF analyses (Appendix A).

### 4.4. Antimicrobial Susceptibility

Antimicrobial susceptibility profile was conducted by Kirby Bauer, following a previously standardized protocol [41]. Based on the NARMS Gram positive panel [42], we tested the following antimicrobials for antimicrobial susceptibility; azithromycin, cefoxitin, ciprofloxacin, clindamycin, chloramphenicol, gentamicin, linezolid, oxacillin , penicillin, rifampicin, trimethoprim/sulfamethoxazole, tetracycline (OXOID™ , Hampshire, UK). We interpreted our results based on the recommendations of the Clinical and Laboratory Standards Institute (CLSI) [43] (Supplementary material B). *Staphylococcus aureus* ATCC 25923 was used as control.

### 4.5. Detection and Confirmation of Methicillin-Resistance, mecA and mecC Genes

All isolates identified by MALDI-TOF were submitted for PCR-testing to identify *mec*A and *mec*C genes following methods previously described [27]. Control strains for *mec*A (SARM 14) and *mec*C (*mec*ALGA251) genes were facilitated by Dr. Gerardo Gonzalez (LIAA, Concepción, Chile) and Dr. Rhod Larsen (National Reference Laboratory for Staphylococci, Staten Serum Institute, Copenhagen, Denmark) (Appendix A in Appendix A). Since confirmation of MRSA being clinically relevant, it led us to further analysis, which consisted in performing additional PCR testing according to Stegger et al. [27], and the amplicon was sequenced with Sanger technologies in MACROGEN™ (Korea), considering three isolates. BLASTn was used to compare the sequences [44]. Subsequently, the percentage of nucleotide identity between them was evaluated using Clustal Omega 2.1 [45] (Appendix A in Appendix A).

### 4.6. Subtyping by PFGE of Strains of S. aureus

Molecular typing of MRSA was performed by genome macro-restriction followed by pulsed field gel electrophoresis (PFGE) in a CHEF DR-II apparatus (Bio-Rad, La Jolla, CA) according to McDougal et al. [46], and interpreted according to Tenovers criteria [47]. *Salmonella enterica* subsp. *enterica* serotype Braenderup H9812 strain was used as a DNA molecular size control (Appendix A).

### 4.7. Statistical Analysis

We employed four univariate and multivariate analyses to test whether a subset of independent variables was associated with antimicrobial resistance. Antimicrobial resistance was defined as resistance to at least one of the antibiotics tested. The four models included were: (I) MRCoNS, (II) MRSA, (III) either MRCoNS or MRSA, (IV) MRCoNS and MRSA strains simultaneously. We computed logistic regressions using robust standard errors to look at the change in the odds of resistance levels (Appendix A). We firstly computed univariate logistic regression models, and those independent variables that were statistically significant at *p*-value < 0.1, in most four models, were incorporated in the multivariate logistic analysis [48]. We also employed Wald and Likelihood ratio tests to select and evaluate our independent variables, which supported our selection of variables from the univariate regression results [49]. Odds are reported as odds ratios (OR) with their respective 95% confidence intervals (CI) and *p*-values. We used a 5% cut off point for the significance level in our multivariate analysis (*p*-value < 0.05). In the multivariate analysis, reference groups comprised non-significant subcategories within each itemized variable from the univariate analysis (e.g., non-significant season’s types were grouped as a zero value, whereas the significant season’s category was coded as one).

## 5. Conclusions

The study results demonstrate the presence of MRSA and MRCoNS isolates in the environment and in equine patients at a Chilean veterinary hospital. Our findings are informative of the risks of potential transmission produced by the contact between human (healthcare workers) and animal-patients. Effective interventions should take place at the community level to control the spread of these pathogens through intensive screening and sanitation protocols, but also at the hospital level by raising awareness and hygienic decolonization measures. From the perspective of One Health, guidelines and policy protocols aiming to employ a wider control over antimicrobial consumption and exposure to AMR reservoirs in animals should be advocated.

## Figures and Tables

**Figure 1 antibiotics-11-00621-f001:**
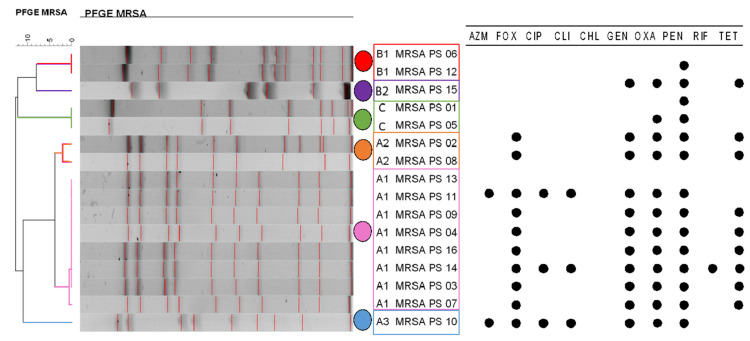
Dendrogram representation of MRSA isolates clustered using the UPGMA method. Six pulse types of MRSA are identified at the right, colors have been assigned for each pulse type (A–C), matching Figure 2B). On the far right, there is a representation of the susceptibility profile of the isolates. AZM: azithromycin; FOX: cefoxitin; CIP: ciprofloxacin; CLI; clindamycin; CHL: chloramphenicol; GEN: gentamicin; OXA: oxacillin; PEN: penicillin; RIF: rifampicin; TET: tetracycline.

**Figure 2 antibiotics-11-00621-f002:**
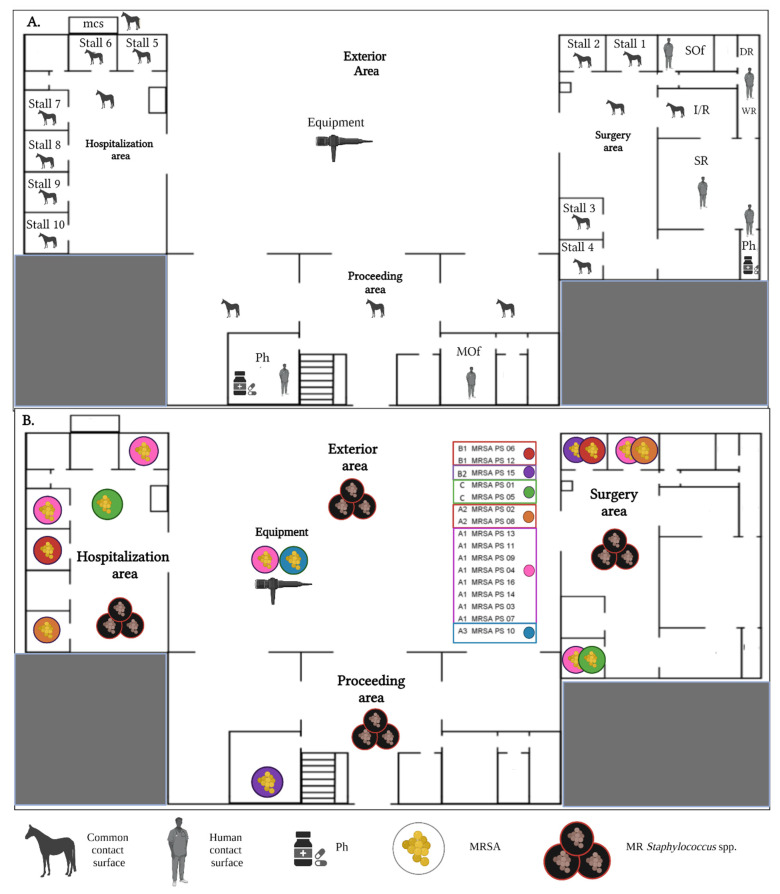
(**A**). Diagram of the space areas within the Equine Veterinary Hospital. Use of areas considering the movement of animals and humans; (**B**). Locations *Staphylococcus* strains isolation and in colors pulse types of MRSA found in this study (colors matching Figure 1). Mcs: manure collection site; I/R: induction recovery room; SR: surgery room; SOf: surgery office; MOf: main office; Ph: Pharmacy, DR: dressing room; WR: washing room.

**Table 1 antibiotics-11-00621-t001:** Results of methicillin resistant *Staphylococcus aureus* (MRSA) and methicillin resistant coagulase-negative staphylococci (MRCoNS) surveillance collected in the Equine Veterinary Hospital during the study.

Sampling Number	Month/Season		Number of Samples	Number of Samples from Equines (Number of Equines (1))	Number of Environmental Samples (2)	Nº of MRSA	Nº of MRCoNS
1	July	Winter 2015	82	21 (5)	61	2	13
2	August		82	21 (5)	61		
3	September	Spring 2015	90	29 (7)	61		
4	October		75	14 (3)	61	6	7
5	November		86	25 (6)	61		
6	December	Summer 2015–2016	92	31 (7)	61		
7	January		99	38 (9)	61	3	8
8	February		71	10 (2)	61		
9	March	Autumn 2016	81	20 (5)	61		
10	April		71	10 (1)	61	5	8
11	May		78	17 (4)	61		
12	June	Winter 2016	71	10 (1)	61		
Total			978	246 (55)	732	16	36

(1) Number of equine patients. (2) Environmental sampling was divided into the Exterior, Equipment, Proceedings, Surgery and Hospitalization areas.

**Table 2 antibiotics-11-00621-t002:** Descriptive characteristics of isolated methicillin resistant *Staphylococcus* aureus (MRSA), results are ordered by date of isolation.

Isolation Date	Source (a)	Area (a)	*mecA*	*mecC*	PFGE	Antibiotic Resistance Profiles (f)
July-15	Patient (b)	Surgery	Yes	No	B2	PEN
Augest-15	Patient (b)	Hospitalization	Yes	Yes	A2	FOX - GEN - OXA - PEN - TET
September-15	Patient (b)	Surgery	Yes	Yes	A1	FOX - GEN - OXA - PEN- TET
September-15	Patient (c)	Surgery	Yes	Yes	A1	FOX - GEN - OXA - PEN - TET
October-15	Environmental	Hospitalization (e)	Yes	No	B2	PEN-OXA
October-15	Patient (c)	Surgery	Yes	Yes	B1	Pan-susceptible
October-15	Patient (c)	Surgery	Yes	Yes	A1	FOX - GEN - OXA - PEN - TET
October-15	Environmental	Surgery (e)	Yes	Yes	A2	FOX - GEN - OXA - PEN - TET
January-16	Patient (c)	Surgery	Yes	Yes	A1	FOX - GEN - OXA - PEN - TET
February-16	Environmental	Equipment (e)	Yes	Yes	A3	AZM - FOX - CIP - CLI - OXA - PEN
February-16	Environmental	Equipment (e)	Yes	Yes	A1	AZM - FOX - CIP - CLI - GEN - OXA - PEN - TET
April-16	Patient (c)	Hospitalization	Yes	Yes	B1	FOX - GEN - OXA - PEN - TET
May-16	Environmental	Hospitalization (e)	Yes	No	A1	Pan-susceptible
June-16	Environmental	Hospitalization (e)	Yes	Yes	A1	FOX - GEN - OXA - PEN - TET
June-16	Environmental	Proceeding (d)	Yes	Yes	C	FOX - CLI - GEN - OXA - PEN -TET
June-16	Environmental	Equipment (f)	Yes	Yes	A1	FOX - GEN - OXA - PEN - TET
			16/16 (100)	14/16 (87.5)		

(a) Sources and areas in the hospital where the samples were taken (See Figure 1); (b) Colonized Patient; (c) Clinical Patients; (d) human contact surfaces; (e) common contact surfaces; (f) Abbreviations: azithromycin (AZM), cefoxitin (FOX), ciprofloxacin (CIP), clindamycin (CLI), chloramphenicol (CHL), gentamicin (GEN), linezolid (LZD), oxacillin (OXA), penicillin (PEN), kanamycin (KAN), rifampicin (RIF), trimethoprim/sulfamethoxazole (SXT) and tetracycline (TET).

**Table 3 antibiotics-11-00621-t003:** Descriptive characteristics of isolated methicillin resistant coagulase negative staphylococci (MRCoNS), results are ordered by date of isolation.

Isolation Date	Specie (a)	Source (b,c)	Area (d, e)	*mecA*	*mecC*	Antibiotic Resistance Profiles (f)
July-15	*S. vitulinus*	Patient (b)	Surgery	Yes	Yes	Pan-susceptible
July-15	*S. sciuri*	Patient (b)	Hospitalization	Yes	Yes	OXA - PEN
July-15	*S. sciuri*	Patient (b)	Hospitalization	Yes	Yes	AZM - FOX - CIP - OXA - PEN - TET
July-15	*S. sciuri*	Environmental	Surgery (e)	Yes	Yes	FOX - GEN - OXA - PEN
July-15	*S. vitulinus*	Environmental	Proceeding (e)	Yes	Yes	CLI - OXA
July-15	*S. sciuri*	Environmental	Proceeding (e)	Yes	Yes	FOX - GEN - OXA - PEN
July-15	*S. vitulinus*	Environmental	Surgery (e)	Yes	Yes	OXA
July-15	*S. sciuri*	Environmental	Surgery (e)	Yes	Yes	FOX - CLI - OXA - PEN
July-15	CoNS	Environmental	Exterior (e)	Yes	No	FOX - CLI - OXA - PEN
July-15	*S. sciuri*	Patient (b)	Hospitalization	Yes	No	FOX - GEN - OXA - PEN
Ago-15	*S. sciuri*	Patient (b)	Hospitalization	Yes	No	FOX - CLI - GEN - OXA - PEN
Ago-15	*S. lentus*	Patient (b)	Surgery	Yes	Yes	FOX - CIP - CLI - CHL - GEN - OXA - PEN - SXT
Ago-15	*S. sciuri*	Patient (b)	Surgery	Yes	Yes	FOX - GEN - OXA - PEN
October-15	CoNS	Environmental	Proceeding (e)	Yes	Yes	FOX - CLI - OXA - PEN
October-15	CoNS	Patient (b)	Surgery	Yes	Yes	FOX - CIP - CLI - OXA - PEN
October-15	*S. sciuri*	Environmental	Exterior (e)	No	Yes	FOX - CLI - OXA - PEN
October-15	CoNS	Environmental	Exterior (e)	Yes	Yes	Pan-susceptible
October-15	*S. sciuri*	Environmental	Equipment (e)	Yes	No	CLI - OXA
October-15	*S. vitulinus*	Environmental	Equipment (e)	No	Yes	OXA - PEN
October-15	CoNS	Environmental	Hospitalization (d)	Yes	Yes	FOX - GEN - OXA - PEN
January-16	*S. sciuri*	Patient (b)	Surgery	Yes	Yes	FOX - CLI - GEN - OXA - PEN
January-16	*S. sciuri*	Patient (b)	Surgery	Yes	Yes	AZM - FOX - CIP - CLI - CHL - GEN - OXA - PEN - RIF - TET
January-16	*S. sciuri*	Patient (b)	Hospitalization	Yes	Yes	FOX - GEN - OXA - PEN
January-16	CoNS	Environmental	Proceeding (e)	Yes	Yes	OXA
January-16	*S. sciuri*	Environmental	Exterior (e)	Yes	Yes	FOX - OXA - PEN
February-16	*S. sciuri*	Environmental	Equipment (e)	Yes	Yes	AZM - FOX - CIP - CLI - OXA - PEN
February-16	*S. sciuri*	Environmental	Equipment (e)	Yes	Yes	FOX - CHL - OXA - PEN
February-16	CoNS	Environmental	Equipment (e)	Yes	Yes	AZM - FOX - CIP - CLI - GEN - OXA - PEN
March-16	*S. sciuri*	Environmental	Equipment (e)	Yes	Yes	FOX - CHL - GEN - OXA - PEN
March-16	*S. sciuri*	Environmental	Surgery (e)	Yes	Yes	AZM - FOX - CIP - CLI - CHL - OXA - PEN - SXT
March-16	*S. sciuri*	Patient (b)	Surgery	Yes	No	AZM - FOX - CIP - CLI - GEN - OXA - PEN - TET
March-16	CoNS	Patient (b)	Hospitalization	Yes	Yes	RIF - TET
May-16	CoNS	Environmental	Hospitalization (d)	Yes	Yes	Pan-susceptible
June-16	*S. sciuri*	Environmental	Hospitalization (d)	Yes	Yes	FOX - CLI - OXA - PEN
June-16	CoNS	Environmental	Proceeding (e)	Yes	Yes	FOX - CLI - GEN - OXA -PEN - RIF
June-16	CoNS	Patient (b)	Surgery	Yes	Yes	FOX
				34/36 (94%)	31/36 (86%)	

(a) Species identification of these samples was performed by MAILDI-TOF. (b) Colonized Patient; (c) Clinical Patients; (d) common contact surfaces; (e) human contact surfaces; (f) Abbreviations: azithromycin (AZM), cefoxitin (FOX), ciprofloxacin (CIP), clindamycin (CLI), chloramphenicol (CHL), gentamycin (GEN), linezolid (LZD), oxacillin (OXA), penicillin (PEN), kanamycin (KAN), rifampicin (RIF), trimethoprim/sulfamethoxazole (SXT) and tetracycline (TET).

## Data Availability

The data presented in this study are available in Appendix A.

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
