# Peer review of "Mec-Positive Staphylococcus Healthcare-Associated Infections Presenting High Transmission Risks for Antimicrobial-Resistant Strains in an Equine Hospital"

_antibiotics, 2022, doi:10.3390/antibiotics11050621_

Round 1

Reviewer 1 Report

In general, this work is really interesting and well written. My comments aim to increase the scientific soundness and clarity of it.

  1. Abstract – the aim of the study should end with the information how the authors intend to accomplish it (brief description of methods used are missing).
  2. Figure 2 – what DR, WR and SR stand for ?
  3. Chapter 4.4. – All mentioned antimicrobials have been already abbreviated (see chapter 2.2.) There is no need to abbreviate it for the second time.
  4. Chapter 4.7 – please specify exactly what kind of statistical test(s) were used. Were any post-hoc test applied?
  5. Please add the data presentation format.
  6. It is not clear what level of probability was considered statistically significant or statistically highly significant.
  7. The authors should once again checked the correctness of references. Some examples of minor errors:
  • Reference 1 – please provide names of all authors
  • Reference 16 – please delete one coma in “Magiorakos, A..;”
  • Reference 21 – please correct the first name initial in  “Shimizu, a;”
  • Reference 26 – as above

Author Response

Reviewer 1

  1. Abstract – the aim of the study should end with the information how the authors intend to accomplish it (brief description of methods used are missing).

Response: We have added the aim of the study into the abstract, and also some extra methodological details.

  1. Figure 2 – what DR, WR and SR stand for?

Response: Thank you for noticing. DR: dressing room; SR: surgery room; WR: washing room. Already added to the figure legend.

  1. Chapter 4.4. – All mentioned antimicrobials have been already abbreviated (see chapter 2.2.) There is no need to abbreviate it for the second time.

Response: All the abbreviations in chapter 4.4 were deleted, only full names were left.

  1. Chapter 4.7 – please specify exactly what kind of statistical test(s) were used. Were any post-hoc test applied?

Response: We first estimated univariate models (Appendix A, Table A2) and those variables having a statistically significant estimate value were selected as candidates for the multivariate analysis. We base this on the results of the univariate logistic regression model, according to a p-value cut-off point of 0.1. All independent variables fulfilling the cut-off point criteria in most of the four models performed were incorporated in our multivariate analysis (Appendix A, Table A3). This has been clarified throughout the statistical analysis section. Also, we employed Wald and Likelihood ratio test for all possible estimated coefficients and models. All the statistically significant variables in the univariate model resulted in a statistically significant improvement in model fit according to the Likelihood ratio tests (Chi2>4, prob>Chi2=<0.001). Wald test provided similar outputs and suggested that all non-significant variables' coefficients at the univariate logistic regression were very small relative to its standard errors; therefore, removing them from a multivariate model did not reduce the model's overall fit. We added a footnote explaining this under Table A2. All the methods selected are based on previous literature and are described in detail elsewhere 1. Finally, we did not apply any posthoc tests (e.g., ANOVA) because the univariate regression models identify similar patterns (especially for linear regressions). Still, we think the univariate logistic regression is in line with the distribution of our binary response variable and is preferred. It adjusts and performs better than other approaches when examining the association between a categorical dependent variable and continuous/categorical independent variables 2.

  1. Please add the data presentation format.

Response: We thank the reviewer for the comment. This has been clarified thoroughly in the methods section, under the statistical analysis’ subsection.

  1. It is not clear what level of probability was considered statistically significant or statistically highly significant.

Response: We used the most conservative significance level of 10% given the small sample size and potential low representativeness given its observational design —we could not perform a randomized controlled trial—. Nevertheless, we also reported p-values, so the reader might be able to analyze the extent to which our coefficients are significant or non-significant using different levels of significance or cut-off points (e.g., 5%, 1%).

This has been clarified accordingly in the methods section.

  1. The authors should once again checked the correctness of references. Some examples of minor errors:

Response: Thank you very much for pointing that out. It is already corrected. We have double checked the other references.

  • Reference 1 – please provide names of all authors
  • Reference 16 – please delete one coma in “Magiorakos, A..;”
  • Reference 21 – please correct the first name initial in  “Shimizu, a;”
  • Reference 26 – as above

Reviewer 2 Report

Data on AMR of bacteria isolated from patients in veterinary hospitals are interesting; however, I suggest the authors consider the following improvements:

-       Title: Latin names do not declension

-       Is number 16 MRSA from 978 samples high as written in the conclusions?

-       The bacterial number from environmental samples is much higher than the bacterial number from patients. How was this considered during data interpretation?

-       The mecA and mecC sequences should be checked for all isolates, not only for three.

-       Do you have some indication for the following statement in discussion: »This could be explained by regulatory genes which suppress mec genes - and subsequently encoding of PBP2a - in the absence of β-lactam antibiotics [26].« Did you analyze sequences?

-       Could you name the cluster mentioned in 3.4.? I am not sure if the calling cluster is here the most appropriate. Do you want to say that strains with the same AMR and other characteristics were isolated from

Author Response

Data on AMR of bacteria isolated from patients in veterinary hospitals are interesting; however, I suggest the authors consider the following improvements:

  1. Title: Latin names do not declension

Response: Thank you for your suggestion. The declension was deleted.

  1. Is number 16 MRSA from 978 samples high as written in the conclusions?

Response: We acknowledge this comment, we have deleted “high prevalence” in the conclusion to better reflect our results.

  1. The bacterial number from environmental samples is much higher than the bacterial number from patients. How was this considered during data interpretation?

Response: We have incorporated an appropriate interpretation on the excess bacterial burden among environmental samples, compared to patients’, throughout the discussion.

  1. The mecA and mecC sequences should be checked for all isolates, not only for three.

Response: Thank you so much for this comment. This project has little funding, then we needed to select only a sample of all mecA and mecC positives. We have added a sentence in the text highlighting this limitation. 

  1. Do you have some indication for the following statement in discussion: »This could be explained by regulatory genes which suppress mec genes - and subsequently encoding of PBP2a - in the absence of β-lactam antibiotics [26].« Did you analyze sequences?

Response: We sequenced a 729 bp fragment, which did not include the regulatory genes. Therefore, we are discussing a possible explanation for the susceptible strains with positive PCR for mec. We added a sentence mentioning the need to further analyze the regulatory genes.

  1. Could you name the cluster mentioned in 3.4.? I am not sure if the calling cluster is here the most appropriate. Do you want to say that strains with the same AMR and other characteristics were isolated from

Response: Thank you very much for your suggestion. We have revised this to name it as before (Pulse-type).